# Biosensors for the Detection of Enzymes Based on Aggregation-Induced Emission

**DOI:** 10.3390/bios12110953

**Published:** 2022-11-01

**Authors:** Fengli Gao, Gang Liu, Mingyi Qiao, Yingying Li, Xinyao Yi

**Affiliations:** 1College of Chemistry and Chemical Engineering, Anyang Normal University, Anyang 455000, China; 2College of Chemistry and Chemical Engineering, Central South University, Changsha 410083, China

**Keywords:** aggregation-induced emission, enzymes, fluorescent biosensors, organic dyes, metal nanoclusters

## Abstract

Enzymes play a critical role in most complex biochemical processes. Some of them can be regarded as biomarkers for disease diagnosis. Taking advantage of aggregation-induced emission (AIE)-based biosensors, a series of fluorogens with AIE characteristics (AIEgens) have been designed and synthesized for the detection and imaging of enzymes. In this work, we summarized the advances in AIEgens-based probes and sensing platforms for the fluorescent detection of enzymes, including proteases, phosphatases, glycosidases, cholinesterases, telomerase and others. The AIEgens involve organic dyes and metal nanoclusters. This work provides valuable references for the design of novel AIE-based sensing platforms.

## 1. Introduction

Enzymes are a family of proteins or RNA with specific catalytic activities toward the substrates. They are involved in most of the complex biochemical processes, including gene expression, metabolic pathways, cell growth and differentiation, and signaling [1]. Abnormal manifestations of enzyme activities could be indicative of humans’ health status. Thus, enzymes can be used as biomarkers for disease diagnosis [2,3,4]. For example, changes in the activity and content of proteases may cause some diseases, such as cancers, acquired immune deficiency syndrome, and neurodegenerative diseases [5,6]. Phosphatases have been considered important indicators for some diseases, including anemia, chronic nephritis, hypothyroidism, and hepatobiliary as well as bone diseases [7]. Evaluated concentrations of γ-glutamyltranspeptidase have been found in hepatocellular carcinoma as well as cervical and ovarian cancers [8]. Excessive use of organophosphorus pesticides can cause the inhibition of acetylcholinesterase (AChE) activity and the accumulation of neurotransmitter acetylcholine, eventually leading to the incidence of Alzheimer’s and Parkinson’s diseases [9]. In acute pancreatitis, lipase concentration in serum is at least three times the normal level [10]. Increased leucine aminopeptidase expression is closely related to many diseases, such as hepatic dysfunction and liver cancer [11].

Owing to the importance of enzymes in clinical diagnosis, various novel methods have been developed for the accurate and sensitive determination of enzymes, such as fluorescence, colorimetry, electrochemistry, electrochemiluminescence, and photoelectrochemistry [12,13]. Among them, fluorescent biosensors have attracted extensive attentions due to their excellent advantages of inexpensive equipment, rapid response, high sensitivity, and real-time and on-site detection [14,15]. However, traditional fluorophores with planar and intermolecular interaction may suffer from the aggregation-caused quenching (ACQ) problem at high concentrations or in the solid state [16], which will decrease the sensitivity and limit the applications of fluorescent biosensors in vivo detection and imaging of enzymes. Since Tang’s group first introduced the concept of aggregation-induced emission (AIE) in 2001, AIE-based techniques have opened up the field with huge potential practical applications [17]. Unlike ACQ fluorophores, fluorogens with AIE features (AIEgens) exhibit a remarkable optical property and strong resistance toward photobleaching. Generally, AIEgens in the molecular state are non-emissive, because they can consume the excitation energy through the active intramolecular motion in a non-radiative decay manner. However, under the restrictions of intermolecular motion, AIEgens in the aggregate state will produce significantly enhanced and stable emission. Thus, versatile AIEgens have been used as functional components to prepare light-up probes. They have been used in a broad range of applications, including organic optoelectronic devices, biosensing, bioimaging, photodynamic therapy, photothermal therapy, and so on [18,19,20,21].

Within the continuous exploration of the new AIE systems, more mechanisms have been proposed as branches of AIE, including crystallization-induced emission, room-temperature phosphorescence, clusterization-triggered emission, and so on [22,23]. Besides tetraphenylsilols and tetraphenylethylene, versatile AIEgens with different structures have been innovatively synthesized for various applications under the guidance of these AIE mechanisms in recent years, such as tetraphenylpyrazines, perylene, organoboron or carborane complexes, cyclooctatetrathiophene, and so on [24,25]. Through modification of the π-conjugation systems with different functional groups, the absorption/emission wavelengths of AIEgens can cover the whole visible and near-infrared range. When the ACQ dyes were modified with AIEgens, the resulting probes may exhibit AIE characteristics. Moreover, some nanomaterials have also been reported to possess AIE properties, including quantum dots (QDs), copper nanoclusters (CuNCs), gold nanoclusters (AuNCs), silver nanoclusters (AgNCs) and carbon quantum dots [26,27]. The aggregation of AIEgens can be triggered by different stimuli, such as solubility change, hydrophobic assembly, hydrogen bonding, target-receptor binding, and electrostatic interactions. By linking AIEgens to the target ligands or reactive groups, many AIE light-up sensors have been developed for the signal-on detection of various targets, including ions, small molecules, microenvironment sensing (e.g., pH, temperature and viscosity), biological macromolecules (e.g., toxin, nucleic acids, proteins, enzymes), cellular processes and pathogens [28,29]. Thereby, the in vitro determination and in vivo monitoring of enzymes have been realized using AIEgens to label peptide probes due to their good biocompatibility, adjustable fluorescence wavelength, ultra-low background signal, and strong photobleaching resistance [30,31]. Moreover, peptides can improve the hydrophilicity of AIEgens, especially in imaging applications. In the presence of target enzymes, AIEgens-labeled peptide probes would be cleaved, causing the aggregation of AIEgens and thus generating a bright AIE luminescence for the in situ imaging of enzymes. Based on the approaches to activate the restriction of intramolecular motion and trigger the AIE process, the detection mechanisms of the works involved in this review could be classified into three categories: (i) the electrostatic interactions between AIEgens themselves or between AIEgens and other charged species (e.g., polymers and nanomaterials) cause the aggregation of AIEgens, (ii) enzyme catalysis adjusts the solubility of AIEgens and triggers the appearance of nanoaggregates, and (iii) enzyme catalysis induces the formation of intramolecular hydrogen bonds that can hamper the restriction of intramolecular motion and produce the effect of excited-state intramolecular proton transfer. These mechanisms have provided versatile approaches for the sensitive and selective detection of enzyme activity in vitro or in vivo.

Some excellent reviews have summarized the progresses in AIE-based applications, which mainly focus on the molecular design, detection mechanism, and AIEgens [32,33,34,35,36]. For instance, Liu et al. reported recent advances in AIE light-up probes for photodynamic therapy [37]. The achievements of tetraphenylethylene (TPE)-based AIE-active sensing probes have been reviewed by Bhosale’s group [32]. Wang et al. summarized the development of Schiff base AIEgens for sensing applications [38]. In this work, we aimed to highlight the advances in AIEgens-based probes and sensing platforms for the detection of enzymes. According to the type of enzymes, AIE-based biosensors have been classified into six categories: proteases, phosphatases, glycosidases, cholinesterases, telomerase and others. We mainly focus on the enzyme-responsive detection mechanisms and the analytical performances.

## 2. Proteases

With AIEgens-labeled peptide as the probe, the fluorescent detection and imaging of proteases has been achieved, including chymase, caspases, thrombin, metalloproteinase-2 (MMP-2), furin, carboxypeptidase Y (CPY), autophagy-related cysteine protease ATG4B, and so on (Table 1). The AIEgens involve the derivatives of TPE, PyTPA, (2-(2′-hydroxyphenyl)-4(3H)-quinazolinone (HPQ), pyrene and tetraphenylsilole (BATPS). Moreover, through the integration of AIE and excited-state intramolecular proton transfer (ESIPT) characteristics, some proteases such as aminopeptidase could be determined with leucine-conjugated TPE or tetraarylimidazole scaffolds [39,40]. In this section, we primarily summarized the AIEgen-labeled peptide probes for the detection and in vivo imaging of proteases. Meanwhile, applications of the AIE probes for monitoring apoptosis and tracking drug delivery were also discussed.

Among the different kinds of AIEgens, TPE is most the commonly used for the modification of protease substrates because of its advantages of easy synthesis and functionalization and excellent AIE characteristics [41]. Zhang et al. designed a fluorescent light-up probe for chymase detection. The probe contains the AIEgen TPE-thiophene (TPETH) with red emission in the aggregation state and two chymase-specific peptide (CFTER) sequences (Figure 1A) [42]. In order to improve the water-solubility and decrease the background, three aspartic acid residues (D_3_) were introduced to the peptide sequence. The probe of TPETH-2(CFTERD_3_) showed nearly no emission in buffer solution. After the enzymatic cleavage, the released hydrophobic TPETH segments could readily assemble into aggregates, lighting up the fluorescence. In order to promote the assembly of TPE and enhance the sensitivity of AIE biosensors, a self-assembly peptide could be incorporated into the probe. For example, Han et al. designed an AIEgen probe using a short self-assembly peptide GFFY to conjugate the protease-responsive peptide and AIEgen residue [43]. After responding to caspase-3, GFFY promoted the ordered assembly of AIEgen residues, substantially restricting the intramolecular motion of AIEgens and greatly improving the sensitivity. In addition, Li et al. designed an AIEgen probe named HPQF for the in situ detection and imaging of endogenous furin (Figure 1B) [44]. The probe consists of a furin-specific peptide (RVRR) and the fluorophore 6-chloro-2-(2-hydroxyphenyl) quinazolin-4(3H)-one (Cl-HPQ). The catalytic cleavage of peptide probes by furin caused the release of insoluble Cl-HPQ precipitates, turning on the fluorescence through the AIE effect.

In order to improve the sensitivity of AIE-based biosensors, Wu et al. developed a two-step sensing system for protease detection based on the peptide probe of RRRRRRGGPLGLAGPra(PyTPA)-NH_2_ (MP), negatively charged nanoparticles (NPs) and slippery lubricant-infused porous substrates (SLIPS) (Figure 2) [45]. The water-soluble MP with an AIEgen of PyTPA showed weak fluorescence. Cleavage of MP by MMP-2 promoted the aggregation of the hydrophobic PyTPA-included residues. The aggregates were then electrostatically adsorbed onto the negatively charged NPs on SLIPS. With MMP-2 as the analytical example, the sensitivity was obviously improved in contrast to the previously reported AIE-based methods.

Real-time monitoring of apoptosis can provide valuable insights for the early detection of therapeutic effects and the evaluation of disease progress. Shi et al. designed an AIEgen probe for the real-time imaging of cell apoptosis by monitoring the caspase activity (Figure 3A) [46]. The probe consists of a hydrophilic caspase-specific peptide (DEVD) and a hydrophobic AIEgen of TPE unit. The water-soluble probe showed almost no fluorescence in the aqueous phase. Cleavage of the probe by caspase-3/7 led to the formation of hydrophobic TPE residues. The aggregation of TPE residues could restrict the intramolecular rotation of the TPE phenyl ring and populate the radiative decay channel, thus lighting up the fluorescence. Moreover, Ding et al. designed an AIEgen probe for the imaging of cell apoptosis by monitoring the activity of caspase-3 [47]. The probe (Ac-DEVD-TPS-cRGD) consists of a caspase-specific hydrophilic peptide (DEVD), a cell-binding cyclic peptide cRGD and an AIEgen of a tetraphenylsilole (TPS) unit. The two peptides were linked at both ends of the TPS unit. Cleavage of the probe by caspase-3 led to the release of TPS-cRGD. The aggregation of released TPS-cRGD residues lit up the fluorescence, enabling real-time imaging of cell apoptosis. Lately, Yuan et al. reported a dual-signal AIEgen probe for monitoring caspase activity. The probe (TPETH–DVEDIETD–TPS) includes three components: a hydrophilic peptide (DVEDIETD) specific to apoptosis initiator caspase-8 and effector caspase-3 and two AIEgens with green (TPS) and red (TPETH) emission colors [48]. The probe could be cleaved into a TPETH–DVEDIETD and a TPS unit by caspase-8. The product of TPETH–DVEDIETD could be further cut into a TPETH unit and DVEDIETD by caspase-3. The aggregates of TPS and TPETH emitted green and red fluorescence, respectively.

Effective methods for cancer treatment largely depend on inducing the apoptosis of cancer cells through chemotherapy and/or radiotherapy. Magnetic resonance (MR) imaging is one of the frontiers of experimental and clinical radiology. It has unlimited tissue penetration depth, excellent soft tissue contrast and time–space resolution, without the use of ionizing radiation. Recently, Meade’s group reported a bimodal fluorescence–magnetic resonance (FL-MR) probe named CP1 for apoptosis imaging by monitoring the activity of caspase-3/7 (Figure 3B) [49]. The CP1 probe contains three parts: a DOTA-Gd(III) chelate for MR signal enhancement, an AIEgen of the TPE unit, and an enzyme-specific peptide, DEVD. In the presence of caspase-3/7, the water-soluble peptide DEVD was removed, and the resulting Gd(III)-AIEgen (Gad-AIE) conjugates tend to aggregate, thus inducing the increase of FL-MR signals. The dual-signal FL–MR probe was successfully used for the fluorescent imaging of cell apoptosis.

In addition, Yuan et al. designed a protease-responsive chemotherapeutic Pt(IV) prodrug with real-time in situ monitoring of drug release (Figure 4A) [50]. The chemotherapeutic Pt(IV) prodrug was modified with a cell-binding cyclic tripeptide, cRGD, and a TPS-labeled enzyme-specific peptide, DEVD. Pt(IV) could be reduced in active Pt(II) in cells and release the TPS-DEVD simultaneously. The cell apoptosis induced by Pt(II) activated the activity of caspase-3 to cleave TPS-DEVD. The released TPS residues tend to aggregate, thus causing fluorescence enhancement. Chen et al. designed a protease-responsive prodrug (DOX-FCPP-PyTPE, DFP) with AIE characteristics (Figure 4B) [51]. The drug delivery and release in living cells could be controlled and tracked. The DFP prodrug involved three parts: AIEgen PyTPE, functionalized cell-penetrating peptide (FCPP) with a cell-penetrating peptide (CPP) and a short MMP-2-specific peptide (LGLAG), and a therapeutic drug (doxorubicin, DOX). The prodrug itself cannot enter the cells. However, the prodrug could be cleaved by MMP-2 into two components: DOX-linked CPP and a PyTPE-containing peptide segment. The DOX-linked CPP could go inside the cells through the interaction of CPP with cell membrane, thus realizing controlled drug delivery. The hydrophobic PyTPE-containing peptide segments could self-assemble into aggregates to yield yellow fluorescence, thereby achieving the real-time tracking of drug release. Recently, Qin et al. prepared a self-assembly tracking micelle (TPR@DOX) by modifying the RGD peptide and TPE AIEgen on both ends of a caspase-responsive amphiphilic polymer (Figure 4C) [52]. The drug-induced apoptosis activated caspase to cut the DEVD-containing peptide in the polymer, thus releasing the hydrophobic TPE residues to form fluorescent aggregates. The turn-on of fluorescence indicated the successful delivery and release of DOX.

Autophagy plays a crucial role in the metabolic process. ATG4B is an important autophagy-related protease, which can be used to regulate autophagy for cancer treatment. With a bis(pyrene)-derivative (BP)-labeled peptide (GKGSFGFTG) named DPBP as the AIE probe, Lin et al. proposed an effective strategy for the determination of autophagy in living objects by monitoring the activity of ATG4B (Figure 4D) [53]. The DPBP probe was modified onto the fourth-generation poly(amidoamine) dendrimer (PAMAM) as the carrier. The cleavage of DPBP led to the release of hydrophobic BP residues. The self-assembly of BP residues could enhance the fluorescence by AIE process, thus achieving the determination of autophagy in a living system.

Based on the electrostatic interaction-induced assembly of charged TPE derivatives, several groups have reported the detection of proteases with bovine serum albumin (BSA), protamine and heparin as the triggers. For example, Xu et al. demonstrated that BSA could trigger the assembly of positively charged TPE salts named TPE-2+ through electrostatic interactions, lighting up the fluorescence by AIE process [54]. The digestion of BSA by active trypsin limited the assembly of TPE-2+, thus depressing the fluorescence signal. Jiang’s group reported the fluorescent assay of trypsin through the histone-controlled heparin-triggered aggregation of TPE-2+ by electrostatic interactions (Figure 5A) [55]. The cationic protein of histone could electrostatically bind with the anionic polymer of heparin, hindering the interaction between TPE-2+ and heparin. However, the cleavage of histone by trypsin allowed for the heparin-triggered aggregation of TPE-2+, thus enhancing the emission signal. Moreover, Zhao et al. reported the ratiometric fluorescence sensing of heparin and trypsin with phloxine B (PhB) and TPE-modified electrospun fibrous strips (Figure 5B) [56]. Protamine was attached to the modified fibers via electrostatic interactions, causing the static quenching of PhB fluorescence and the AIE-induced enhancement of TPE emission. The digestion of protamine by heparin or trypsin led to the increase of PhB emission at 574 nm, which was accompanied by the decrease of TPE emission at 472 nm.

With glutathione (GSH)-protected gold nanoclusters (GSH-AuNCs) as the AIEgens and positively charged silicon nanoparticles (SiNPs) as the triggers to induce the assembly of AuNCs, Xue et al. reported a ratiometric fluorescence sensing strategy for trypsin detection (Figure 5C) [57]. Protamine could electrostatically adsorb onto the surface of GSH-AuNCs in order to disperse the assemblies, causing fluorescence quenching at 450 nm. In this process, the emission signal of SiNPs as the internal reference showed a negligible change. Subsequently, the trypsin-catalyzed hydrolysis of protamine dissociated the protamine/GSH-AuNCs complexes, facilitating self-assembly and recovering the fluorescence at 570 nm.

## 3. Phosphatases

Phosphatases widely existing in mammalian body fluids and tissues can catalyze the hydrolysis of phosphate ester during osteogenesis. The abnormal expression of phosphatases is closely related to many diseases, such as liver dysfunction, diabetes, bone tumors, and prostatic cancer [7]. As an important biomarker, alkaline phosphatase (ALP) is one of the most commonly used enzymes in medical diagnosis. Taking advantage of AIE−based biosensors, a series of AIEgens have been designed and synthesized for ALP detection, including phosphorylated TPE (TPE-PA, TPE-2PA and TPE-4PA) [58,59,60], phosphate-modified quinolone–malononitrile (QMTP), and tyrosine phosphate (pY)−included TPEPy-peptide (TPEPy-FpYGpYGpY and TPEPy-^D^F^D^F_P_Y^D^EG^D^K) (Table 2). The AIE detection mechanism is based on the difference in the water solubility between substrates and products. After removing the phosphate group from TPE-phos by ALP−catalytic hydrolysis, the product emitted blue or green color due to the AIE effect (Figure 6A) [59]. This light-up probe could be used for the clinical analysis of ALP with high specificity. The auto-fluorescence from living cells may interfere with AIEgens emitting blue or green color. For this reason, several groups have designed fluorescent probes with red emission for ALP analysis [61,62,63,64]. ESIPT can induce redder emission by altering the conjugation system of fluorophores. For this reason, Song et al. designed a ratiometric strategy for ALP detection with greenish-yellow phosphorylated chalcone derivatives as the probes (Figure 6B) [63]. The enzymatic products could assemble into red fluorescent aggregates with ESIPT and AIE processes, realizing the visual and fluorescent detection of ALP in living cells.

The above AIE probes may suffer from complex design, low sensitivity and limited solubility in 100% aqueous solution. Compared to them, TPE-based peptide probes exhibited weak emission in a solution state and simple structure with easy synthetic modification. In addition, the ordered self-assembly of peptides can limit the intramolecular rotation of AIEgens, thus enhancing the luminescence. In this aspect, AIEgens−conjugated peptides have been used for the design of various AIE-based biosensors [65,66]. For example, Zhang et al. reported the detection of ALP activity in bacteria with a TPEPy-conjugated peptide probe TPEPy-^D^F^D^F_P_Y^D^EG^D^K (Figure 6C) [65]. To enhance the resistance of the probe to proteases, D-amino acids were used instead of natural L-amino acids in the peptide sequence. The dephosphorylated peptide probes could assemble into fibers on the bacterial surface due to their hydrophilicity, thus activating the AIE process and turning on the fluorescence.

Based on the self-assembly of anionic and cationic AIEgens mediated by phosphorylated products via electrostatic interactions, ALP could be determined with low background. For example, Zhao et al. found that the phosphorylation of fluorescein (Flu) on the polyethylene terephthalate (PET) fiber allowed for the attachment of bisquaternary ammonium salt of tetraphenylethylene (TPE-2N^+^) through electrostatic interaction [67]. The assembled TPE-2N^+^ on the phosphorylated PET-Flu fiber (PET-Flu-PO_4_) showed an emission peak at 471 nm. Compared to methods based on the absolute change of the fluorescence intensity, the ratiometric fluorescent assays exhibited minimized environmental fluctuation and enhanced sensitivity. Kaur et al. reported a sensing system for ALP detection involving the AIEgen of di-anionic 1,2-Bis[4-(3-sulfonatopropoxyl)phenyl]-1,2-diphenylethene salt (BSPOTPE) and polycationic protamine sulphate (PrS) [68]. The supramolecular complexes of BSPOTPE-PrS showed strong fluorescence. Hexametaphosphate (HMP) could compete with BSPOTPE to bind PrS, thus leading to the release of BSPOTPE from the supramolecular complexes and a decrease in fluorescence intensity. In contrary, the cleavage of HMP facilitated the formation of supramolecular BSPOTPE-PrS complexes, thus turning on the fluorescence. Moreover, Luo et al. reported the detection of ALP based on the stimulus response–regulated simultaneous ACQ and AIE effects from infinite coordination polymer (ICP) nanoparticles (Figure 7) [69]. The nanoparticles were formed by the self-assembly of 1,1,2,2-tetra(4-carboxylphenyl)ethylene (H_4_TCPE), sulforhodamine 101 (SR101), Cu^2+^ ions, and guanosine-5-monophosphate (GMP). The encapsulation of Cu-GMP made the H_4_TCPE/SR101/Cu-GMP nanoparticles emit blue fluorescence from the AIE of H_4_TCPE at 450 nm, while the ACQ-induced red fluorescence emission of SR101 was inhibited. The enzymatic dephosphorylation of GMP to produce adenosine and PO_4_^−^ destructed the networks of Cu-GMP hosts, leading to the separation of two guests (H_4_TCPE and SR101). Consequently, the AIE of H_4_TCPE with blue fluorescence was quenched, while the ACQ of SR101 with red fluorescence was recovered.

Taking advantage of metal nanoclusters with AIE characteristics, several groups have reported the detection of ALP activity through the phosphorylated product-mediated assembly of CuNCs and AuNCs. Based on the competitive interaction between GSH-CuNCs and PPi with Al^3+^, Geng et al. reported the signal-on detection of ALP [70]. The complexation of Al^3+^ by PPi prevented the Al^3+^-induced aggregation of CuNCs. The enzymatic hydrolysis of PPi allowed for the formation of Al^3+^/CuNCs nanoaggregates, thus activating the AIE process and turning on the fluorescence. Based on a similar sensing mechanism, acid phosphatase was analyzed using penicilamine-capped CuNCs as the AIEgens [71]. The AIE progress of CuNCs was activated by adjusting the pH. The bright red luminescence of the AIE of CuNCs could be quenched by Fe^3+^ ions but not PPi-Fe^3+^ complexes. The enzymatic hydrolysis of PPi released Fe^3+^ ions, thus quenching the fluorescence of CuNCs aggregates. Phenylboronic acids can form boronate ester bonds with the cis-diols of glucose (Glu) molecules. Based on the Glu-triggered assembly of p-mercaptophenylboronic acid, (MBA)-stabilized CuNCs to activate the AIE process. Huang et al. reported the detection of ALP using D-glucose 6-phosphate (P-Glu) to modify the CuNCs (Figure 8A) [72]. In this method, a part of the MBA groups on CuNPs reacted with the cis-diol on P-Glu molecules through the formation of boronate ester bonds. After dephosphorylation, P-Glu was converted into Glu, which triggered the assembly of CuNPs. The method with dual recognitions (ALP/P-Glu and 5,6-diol/MBA) was applied for the in situ imaging of ALP activity in cells.

AuNCs have provided a series of platforms for various fluorescence assays. Han et al. reported a fluorescent and colorimetric dual-signal system for ALP assay through via 2, 6-dichlorophenolindophenol (DCIP)-mediated fluorescence resonance energy transfer (FRET) (Figure 8B) [73]. The positively charged polyallylamine hydrochloride (PAH)-capped AuNCs (PAH-AuNCs) with AIE characteristics showed an emission at 570 nm. The negatively charged DCIP could adsorb on the PAH-AuNCs via electrostatic interaction, thus quenching the fluorescence of PAH-AuNCs by fluorescence resonance energy transfer from PAH-AuNCs to DCIP. L-Ascorbic acid (AA), produced by ALP-catalytic hydrolysis of 2-phospho-L-ascorbic acid (AAP), reduced DCIP and can cause color change from blue to colorless, thereby recovering the quenched fluorescence from PAH-AuNCs. Moreover, it was found that Ce^3+^ but not Ce^4+^ ions could enhance the fluorescence of GSH-protected AuNCs via the AIE process [74]. The enzymatic product of AA could reduce Ce^4+^ into Ce^3+^, thus turning on the fluorescence and achieving the detection of acid phosphatase.

Inorganic pyrophosphatase (PPase) can catalyze the conversion of pyrophosphate (PPi) into phosphate (Pi) ions. It plays an important role in lipid synthesis and decomposition, calcium absorption, bone formation and DNA synthesis, and other biochemical transformations. With glutathione-capped silver nanoclusters (AgNCs) as the AIEgens, Tang et al. reported the assays of PPase by Al^3+^-triggered luminescence switch (Figure 8C) [75]. The Al^3+^/AgNCs aggregates showed bright red luminescence. Complexation of Al^3+^ by PPi but not Pi prevented the aggregation of AgNCs, while the decomposition of PPi by PPase facilitated the formation of Al^3+^/AgNCs aggregates. Moreover, the AIE effect of AgNCs is dependent upon the change of solution pH and temperature.

## 4. Glycosidases

Glycosidases are responsible for removing the monosaccharide residues from glycoconjugates [2]. They are involved in many biological and pathological processes. For example, β-galactosidase is a key enzyme for monitoring gene transcription and transfection efficiency [77]. It is also an important biomarker of cell aging, ovarian cancer and other pathological processes. Several groups have reported the detection of β-galactosidase with AIE-active fluorescent probes, such as TPE-ETh-R-GFFY(gal)ERGD, TPE–DCM, HBTTPAG, TPh-PyBz-βgal, SA-βGal, and QM–βgal (Table 3) [78,79,80,81,82]. The enzymatic hydrolysis of the hydrophilic substrates could cause the release of hydrophobic fluorophores (e.g., TPE, SA or QM moieties), thus facilitating the formation of AIE aggregates with intensive emission. Some of the probes have been used for the on-site sensing and long-term imaging of β-galactosidase in living cells with low background interference. Moreover, the AIE luminogens (AIEgens) have been synthesized and used for the detection of other glycosidases. Typically, α-amylase can hydrolyze the α-1,4-glycosidic bonds in starch for the production of maltose, glucose, syrup, beer, rice wine, soy sauce, vinegar, juice, and so on. It can also be regarded as the biomarker for psychological stress (e.g., eustress, distress, anxiety, and depression) [83]. Shi et al. developed an AIE-based method for the assay of α-amylase with water-soluble TPE-labeled maltotriose as the probe (Figure 9A) [84]. The enzymatic cleavage of α-1,4 glycosidic bonds by α-amylase led to the release of maltotriose units and the production of insoluble TPE residues. Based on the AIE of TPE residues, the activity of α-amylase in a patient’s body fluid has been determined.

**Table 3 biosensors-12-00953-t003:** Analytical performances of AIE-based methods for the detection of glycosidases.

Targets	Probes	Linear Range	LOD	Ref.
β-gal	TPh-PyBz-βgal	0.2~2 U/mL	0.22 U/mL	[78]
β-gal	HBTTPAG	0~3.15 U/mL	3.7 mU/mL	[79]
β-gal	SA-βgal	0~0.1 U/mL	14 mU/mL	[80]
β-gal	QM-βgal	0~6 U	1 mU/mL	[81]
β-gal	TPE-DCM	0~7 U/mL	1.5 U/mL	[82]
α-amylase	TPE-maltotriose	0~45.5 U/L	0.14 U/L	[84]
β-gal	CuNCs	0~200 U/L	0.9 U/L	[85]
β-gal	CuNCs	2.3~96 U/L	0.7 U/L	[86]
β-gal	DTE/β-CD CuNCs	0~50 U/L	0.56 U/L	[87]
GUS	BTBP-Gluc	0~7 U/mL	–	[88]

Abbreviations: β-gal, β-galactosidase; SA, salicylaldehyde azines; QM, qnoline-malononitrile; TPE, tetraphenylethene; DCM, dicyanomethylene-4H-pyran; CuNCs, copper nanoclusters; DTE, dithioerythritol; β-CD, β-cyclodextrin; GUS, β-glucuronidase.

In contrast to organic AIEgens, it has been found that some metal nanoclusters show unique AIE properties with longer wavelength emission, which may decrease the interference of fluorescent proteins in biological systems. For example, Zhao et al. found that CuNCs with weak emission could assemble into particles with bright luminescence through hydrophobic interactions [85]. Unlike the thiolated CuNCs, the intense red luminescence of the 4-methylthiophenol protected CuNCs AIE particles and remained stable in neutral and alkaline solution. β-Galactosidase could trigger the substrate hydrolysis to produce galactose and 4-nitrophenol. The resulting 4-nitrophenol could adsorb on the surface of CuNC particles, thus quenching the luminescence and achieving the quantification of β-galactosidase. Qian’s group found that Al^3+^ could induce the aggregation of GSH-capped CuNCs to activate the AIE process [86]. 4-Nitrophenol produced from the enzymatic hydrolysis of 4-nitrophenyl-β-D-galactopyranoside by β-galactosidase quenched of the emission of CuNCs aggregates. The method could be applied for the assay of β-galactosidase activity under physiological condition. However, CuNCs-based assays only show one function, which may limit the detection sensitivity to a certain extent. Recently, Huang et al. developed an on-off-on fluorescent method for the detection of β-galactosidase with multi-functional dithioerythritol (DTE) and β-cyclodextrin (β-CD)-covered CuNCs as the probes (Figure 9B) [87]. Al^3+^ cations caused the aggregation of CuNCs through the formation of Al-O bonds, thus triggering the AIE property. The product of p-nitrophenol from the enzymatic hydrolysis of 4-nitrophenyl-β-D-galactopyranoside could bind with β-CD to quench the fluorescence of aggregated CuNPs through photoelectron transfer. This is the first report integrating AIE and host–guest interaction in one system.

## 5. Cholinesterases

Cholinesterases are a class of key enzymes for biological nerve transmission. Typically, AChE can degrade acetylcholine between cholinergic synapses, prevent the excitatory effect of neurotransmitters on the postsynaptic membrane, and ensure the normal transmission of neural signals in the organism [9]. Organophosphorus pesticides (OPs) are widely used in agricultural production because of their advantages of effectively controlling pests and improving crop yield. Unfortunately, the abuse of OPs has caused the pollution of water resources, fruits, vegetables and processed food, thus causing great damage to the ecological environment and human health. Serine on AChE can be phosphorylated by OPs, thus inhibiting the enzyme activity and preventing the decomposition of neurotransmitters. Based on the AChE substrate or product-induced aggregation of AIEgens, three TPE derivatives have been designed and used for the evaluation of AChE activity and the detection of OPs content (Table 4), including sulfonated TPE, maleimide-functionalized TPE (TPE-M) and leucine–conjugated TPE (TPE-Leu). In the first report, myristoylcholine was used as the AChE substrate, which can assemble into an aggregate or heteroaggregation complex in the presence of sulfonated TPE through electrostatic interactions (Figure 10A) [89]. The formation of TPE assemblies turned on the fluorescence. However, the AChE-catalytic hydrolysis of myristoylcholine limited the formation of assembles, thus causing the decrease in the fluorescence signal. The method could be used for screening of AChE inhibitors in a signal-on detection format. In the second work, acetylthiocholine (ATCh) was used as the AChE substrate [90]. The hydrolysis product of thiocholine could react with the maleimide ring of TPE-M to form an AIE molecule named TPE-M-S. The resulting TPE-M-S on a paper-based fluorescent sensor significantly enhanced the fluorescence. Additionally, a pH-responsive AIE probe of TPE-Leu has also been used for the assay of AChE [91]. The protonated TPE-Leu was water-soluble under basic conditions, showing poor fluorescence. However, the AIEgen would exhibit strong fluorescence in an acidic environment due to its hydrophobic properties. The AChE-catalytic hydrolysis of acetylcholine into choline and acetic acid decreased the pH value of the solution, thus promoting the formation of TPE-Leu aggregates and turning on the fluorescence.

Gold nanoparticles (AuNPs) show excellent fluorescence quenching efficiency. By integrating AIE with nanotechnology, AChE has been detected with AIE nanoparticles and AuNPs as the sensing platforms [92]. The AIE-Au nanoconjugates showed weak fluorescence due to the fluorescence resonance energy transfer from AIE (QAU-1) nanoparticles to AuNPs. The hydrolysis product of thiocholine could interact with AuNPs by the Au-S interaction to separate AIE nanoparticles from the nanoconjugates, thus recovering the fluorescence. Cai et al. reported a metal−organic framework (MOF)-based system for the detection of AChE with AuNCs as the AIEgens (Figure 10B) [93]. The encapsulation of AuNCs on the ZIF-8 MOF to form AuNCs@ZIF-8 nanocomposites triggered the AIE effect and turned on the fluorescence. In the presence of choline oxidase (CHO), H_2_O_2_ was produced during the enzymatic oxidation of choline. The produced H_2_O_2_ decomposed ZIF-8 and destroyed the AuNCs aggregates on MOF, thus reducing the fluorescence. Moreover, the released AuNCs could be used as peroxidase mimics to catalyze the oxidization of 3,3′,5,5′-tetramethylbenzidine (TMB), thus achieving the visual detection of AChE activity in parallel.

**Figure 10 biosensors-12-00953-f010:**
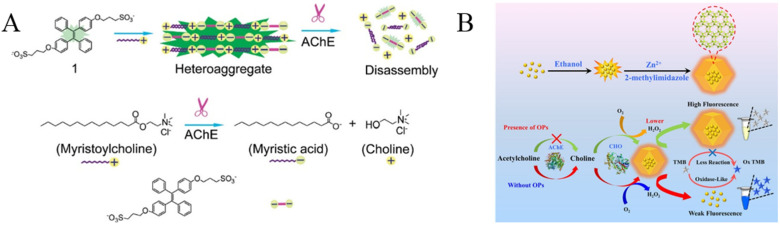
(**A**) Illustration of the formation of heteroaggregate between myristoylcholine and tetraphenylethylene 1 and the disassembly of the aggregate in the presence of AChE. Reprinted with permission from ref. [89]. Copyright 2009, American Chemical Society. (**B**) Schematic diagram of the mechanism for the detection of OPs. Reprinted with permission from ref. [93]. Copyright 2021, American Chemical Society.

Butyrylcholinesterase (BChE) is an important cholinesterase in the process of metabolism and regulation. It is associated with various diseases, such as liver disease, diabetes and Alzheimer’s disease [94]. Recently, Xiang et al. designed a ratiometric fluorescent probe (TB-BChE) for imaging of BChE based on AIE mechanisms [95]. The TB-BChE probe showed a low intramolecular charge transfer effect because of the poor electron-donating ability of the ester group. Once the cyclopropyl group in TB-BChE was removed by BChE, the resulting tricyanofuranyl iminosalicylaldehyde (TCFIS) showed strong intramolecular charge transfer and caused the blue-shifted enhanced fluorescence. The probe was further employed for the ratiometric imaging of endogenous BChE in a nonalcoholic fatty liver mouse model.

**Table 4 biosensors-12-00953-t004:** Analytical performances of AIE-based methods for the detection of cholinesterases.

Targets	Probes	Linear Range	LOD	Ref.
AChE	Sulfonated TPE	0.5~2 U/mL	0.5 U/mL	[89]
AChE	TPE-Maleimide	0.3~3 mU/mL	2.5 mU/mL	[90]
AChE	TPE-Leu	0~100 mU	–	[91]
AChE	AIE-AuNPs	0~8 mU/mL	0.015 mU/mL	[92]
BChE	TB-BChE	0~70 μg/mL	39.24 ng/mL	[95]

Abbreviations: AChE, acetylcholinesterase; TPE, tetraphenylethylene; Leu, leucine; AuNPs, gold nanoparticles; BChE, butyrylcholinesterase.

## 6. Telomerase

Telomerase is a basic nuclear protein reverse transcriptase responsible for telomere elongation in cells [3]. It can add telomere DNA (TTAGGG)_n_ to the chromosome ends of eukaryotic cells, fill in the telomeres lost in DNA replication, and extend telomere repair. Telomerase plays an important role in maintaining chromosome stability and cell activity in different kinds of cells. Its activity in normal human tissues is inhibited, but it can be reactivated in tumors and participates in malignant transformation. Therefore, telomerase is believed to be a tumor biomarker for the early diagnosis, treatment and monitoring of cancers. Based on the electrostatic interactions between positively charged fluorogens and negatively charged extended DNA sequences, Lou and co-workers developed a series of AIE-based sensing systems for telomerase detection [96,97,98,99,100]. In their first study, a water-soluble tetraphenylethene salt (TPE-Z) with two positive charges was used as the AIEgen (Figure 11A) [97]. The fluorescence signal of TPE-Z was poor in the presence of telomerase substrate oligonucleotides (TS primers). When the repeated (TTAGGG)_n_ stands were added to the end of TS primers by telomerase, TPE-Z molecules were bound to the extended DNA backbone, resulting in the increase of fluorescence intensity. Based on the same sensing principle, they reported a more specific strategy for telomerase detection using Silole-R as the AIE indicator (Figure 11B) [98]. In contrast to TPE-Z, Silole-R shows a higher fluorescence quantum yield. Thus, the sensitivity and specificity have been remarkably improved for the detection of telomerase extracted from different cell lines.

For the application of fluorescence probes in a complex biological environment, light-emitting dyes with long wavelengths are particularly popular due to their low background interference from optical self-absorption and autofluorescence. To overcome this limitation, Lou’s group proposed an AIE sensing system for the fluorescent analysis and in situ imaging of telomerase in normal and tumor cells with yellow-emissive AIE dyes (Figure 11C) [99]. The AIEgen (TPE-Py) was synthesized by coupling a pyridinium unit to TPE with vinyl functionality. The positively charged TPE-Py could bind with the free TS primer or quencher-labeled primer (QP) via electrostatic interactions. Without the extension reaction, the emission of TPE-Py aggregates in the primer backbone was quenched due to the fluorescence–resonance energy transfer from TPE-Py to quencher. After the extension reaction, TPE-Py aggregates bound to the repeat units of (TTAGGG)_n_ which are relatively far away from the quencher, thus producing a distance-dependent fluorescence emission. Furthermore, Lou’s group reported an AIE-based ratiometric fluorescent probe for telomerase detection by using Cy5-labeled TS primer and AIEgen Silole-R (Figure 11D) [100]. When the Silole-R molecules were attached onto the extended primer to form the aggregates, a strong blue emission was observed, while the red emission from the internal reference (Cys5) had no change. In the contrast to previous methods, the ratiometric probe showed higher reproducibility and positive result rate for urine samples from bladder cancers.

## 7. Others

γ-Glutamyltranspeptidase (GGT) is believed to be a biomarker of hepatocellular carcinoma (HCC). Its imaging is of great significance for the early monitoring of precise medicine and intraoperative navigation. Based on the enzymatic generation of hydrophobic products, GGT has been determined with TPE derivatives functionalized with γ-glutamyl amide residues. The substrates dispersed in aqueous solution exhibited no or a poor fluorescence peak. When the γ-glutamyl amide group was removed through enzymatic cleavage, the hydrophobic TPE derivatives were assembled into aggregates with blue fluorescence. The probe could be used to determine GGT in serum samples and image endogenous GGT in living cells [101]. In addition, Zeng’s group reported a near-infrared fluorescence probe for GGT imaging by combining AIE and ESIPT (Figure 12A) [102]. The glutamic acid-conjugated ABTT, named ABTT-Glu, is water-soluble and showed almost no fluorescence emission. The GGT-mediated cleavage of ABTT-Glu caused the formation of ABTT aggregates via intramolecular hydrogen interactions, thereby enhancing the fluorescence emission. The probe was further used to detect GGT in HepG2 cells for precision medicine in surgery. Moreover, fluorophores with AIE and ESIPT characteristics have been designed and synthesized for the fluorescent sensing of other enzymes such as lactamase esterases (Table 5) [103,104,105]. For example, esterases are a kind of hydrolase enzyme which participate in a variety of biochemical reactions, depending on the receptors and protein structures and functions; several AIEgen probes have been prepared for esterase detection with the combination of AIE and ESIPT effects by the hydrolysis of esters into acids and alcohols (Figure 12B) [104,105].

Additionally, AIEgen probes have been synthesized for the detection and imaging of other enzymes such as lipase [106,107], fucosidase [108], cyclooxygenase-2 (COX-2) [109] and β-N-acetylhexosaminidase (Hex) [110]. For example, lipase plays an important role in industrial catalysis, drug discovery and disease diagnosis; two AIEgens have been designed and used for the detection of lipase, including TPE and benzophenone derivatives [106,107]. It has been documented that COX-2 is an effective biomarker for the early diagnosis of some cancers. It is overexpressed in cancer cells but not in normal tissues. Xie et al. reported an AIEgen with COX-2 binding ability using the modification of rofecoxib (Figure 12C) [109]. The rofecoxib analogues showed good AIE properties due to the introduction of a phenyl ring. β-N-acetylhexosaminidase (Hex) is a kind of glucoside hydrolase which participates in the catalytic release of N-acetylhexose at the non-reducing end of the substrate. Human Hex (HsHex) plays a key role in lysosomal storage disorders, which can mediate the degradation of GM2 ganglioside in neuronal lysosome [111]. The dysfunction of HsHex can lead to severe neurodegenerative lipid storage disorders. Wang et al., for the first time, designed a lysosome-targeting Hex-lighting-up AIE-active probe (GlcNAc-TPE) for Hex detection (Figure 12D) [110]. The Hex-catalyzed hydrolysis of GlcNAc-TPE produced hydrophobic Py-TPE with poor water solubility. The Py-TPE aggregates boosted the bright emission with large Stokes shift and excellent photostability.

Based on the electrostatic interactions, methyltransferases (MTase) and hyaluronidase (HAase) have also been determined by the AIE mechanism. For example, DNA MTase is a type of enzyme involved in the regulation of gene expression. It can lead to aberrant DNA methylation of tumor-suppressor genes, thus becoming highly methylated and transcriptional silenced in some blood cancers. With the combination of AIE and target-initiated template-free DNA polymerization, Nie et al. designed an AIE probe of TPE-Z to detect MTase [112]. The positively charged TPE-Z bound with single-stranded DNA through electrostatic interactions. Then, DNA MTase catalyzed the polymerization of a sequence-specific hairpin DNA, thus turning on the fluorescence through the AIE effect. The method was used to detect DNA MTase activity in serum samples and evaluate the inhibition efficiency of 5-fluorouracil inhibitor. HAase can catalyze the hydrolysis of hyaluronic acid (HA) into small pieces. It can reduce the activity of hyaluronic acid in the body, thus improving the liquid permeability of tissues. HAase is associated with a variety of physiological and pathological processes, including embryogenesis, inflammation, and wound healing. It has been demonstrated that HAase was overexpressed in some cancer patients, such as those with cancers of the bladder, colon, and prostate [113]. Thus, the evaluation of HAase activity has attracted much attention. Li et al. reported the detection of HAase through the HA-triggered assembly of TPE-4N^+^ compounds via electrostatic interactions [114]. The resulting assemblies showed a yellow-greenish emission. The enzymatic digestion of HA was limited the assembly of TPE-4N^+^, causing the fluorescence quenching. Moreover, based on the quenching ability of AuNPs, Wang et al. reported the detection of HAase with AIE-dot-based nanoprobes (AIEDs) [115]. In this method, the negatively charged HA stabilized AuNPs and promoted their assembly on AIEDs to form HA-AuNPs@AIEDs, thus quenching the fluorescence through fluorescence resonance energy transfer. HAase-induced degradation of HA to small pieces caused the dissociation of AuNPs and thus recovered the fluorescence. Moreover, the HA-AuNPs@AIEDs assemblies could specifically recognize the HA receptors on the cells, thus facilitating endogenous detection and in vitro imaging of HAase.

**Figure 12 biosensors-12-00953-f012:**
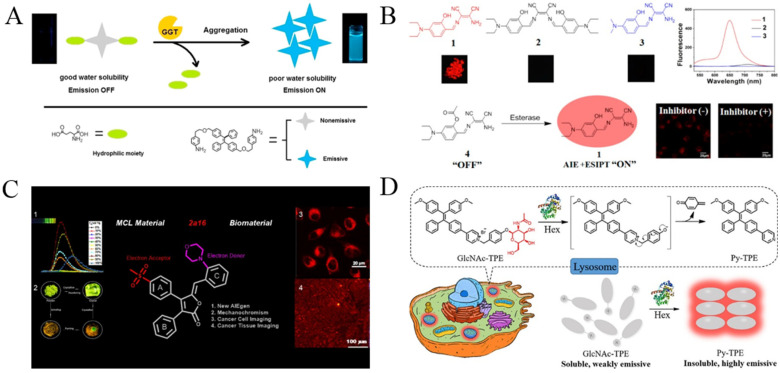
(**A**) Schematic illustration of the probe’s fluorescent detection for GGT. Reprinted with permission from ref. [102]. Copyright 2016, Elsevier. (**B**) Schematic illustration of red-emissive fluorophore with AIE and ESIPT characteristics for light-pp sensing of esterase. Reprinted with permission from ref. [105]. Copyright 2014, American Chemical Society. (**C**) Schematic illustration of rofecoxib-based fluorescent probes for COX-2-targeted bioimaging. Reprinted with permission from ref. [109]. Copyright 2021, American Chemical Society. (**D**) Detection mechanism of GlcNAc-TPE towards Hex. Reprinted with permission from ref. [110]. Copyright 2019, American Chemical Society.

**Table 5 biosensors-12-00953-t005:** Analytical performances of AIE-based methods for the detection of other enzymes.

Targets	Probes	Linear Range	LOD	Ref.
GGT	ABTT-Glu	10~90 U/L	2.9 U/L	[101]
GGT	TPE	0~80 U/L	0.59 U/L	[102]
β-lactamase	DNBS-CSA	0~10 mU/mL	0.5 mU/mL	[103]
Lyso	AIE-Lyso-1	0.1~0.5 U/mL	2.4 mU/mL	[104]
esterase	probe 4	0.01~0.15 U/mL	5 mU/mL	[105]
lipase	TPE−COOC_6_H_13_	0.1~1.3 mg/mL	0.1 mg/mL	[106]
lipase	benzophenone	0.1~4 U/mL	50 mU/mL	[107]
α-fuc	QM-NHαfuc	0~1.75 U/mL	10 mU/mL	[108]
Hex	GlcNAc-TPE	0~0.2 U/mL	3 mU/mL	[110]
MTase	TPE-Z	0.5~100 U/mL	0.16 U/mL	[112]
HAase	TPE-4N^+^	0.05~2 U/mL	20 mU/mL	[114]
HAase	HA-AuNPs@AIEDs	0.01~60 U/mL	7.2 mU/mL	[115]

Abbreviations: GGT, γ-glutamyltranspeptidase; Glu, γ-glutamate; Lyso, lysosomal esterase; CSA, salicylaldehyde azine derivative; α-fuc, α-L-fucosidase; QM, quinoline malononitrile; Hex, β-N-acetylhexosaminidase; MTase, methyltransferases; TPE-Z, tetraphenylethene salt; HAase, hyaluronidase; HA, hyaluronic acid; AuNPs, gold nanoparticles.

## 8. Conclusions

This review has summarized the recent advances in AIEgen-based biosensors for the detection and imaging of enzymes. When triggered by certain stimulus, the well-designed AIEgens in aggregate state exhibit the unique advantages of excellent signal-to-noise ratio, strong photostability and large Stokes’ shift. By simply adjusting the molecule structures, the optical properties of AIEgens can be easily modulated to meet the requirement of enzyme sensing. By selecting the corresponding substrate probes to conjugate with AIEgens, different enzymes have been sensitively detected in vitro and in vivo, including proteases, phosphatases, β-galactosidases, cholinesterases, telomerase, and others. However, there are still some shortcomings and challenges in the AIE-based detection and imaging of enzymes. For example, in order to decrease the back fluorescence from proteins in cells, AIEgens with far-red or near-infrared emission and multiphoton excitation characteristics should be extensively explored for the in vivo sensing and monitoring of enzymatic processes. Second, more detection mechanisms should be proposed in order to detect other types of enzymes, such as redox enzymes and isomerases. Third, aiming to monitor the dynamic enzymatic process, it is necessary to explore reversible enzyme-responsive AIE probes. Lastly, the undefined degradation and cell toxicity of AIEgens-based probes in cells and tissues needs more detailed assessments.

## Figures and Tables

**Figure 1 biosensors-12-00953-f001:**
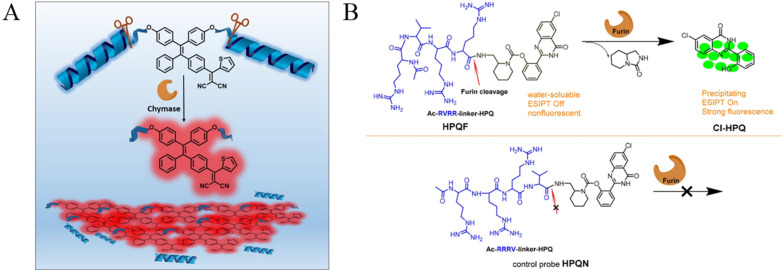
(**A**) Schematic illustration of the functional mechanism of the probe TPETH-2(CFTERD_n_). n = 2 or 3 for chymase sensing. Reprinted with permission from ref. [42]. Copyright 2016, American Chemical Society. (**B**) Chemical structure of the activatable probe HPQF and its working mechanism with furin (top), and the molecular design strategy of the negative control non-cleavable probe HPQN (bottom). Reprinted with permission from ref. [44]. Copyright 2018, American Chemical Society.

**Figure 2 biosensors-12-00953-f002:**
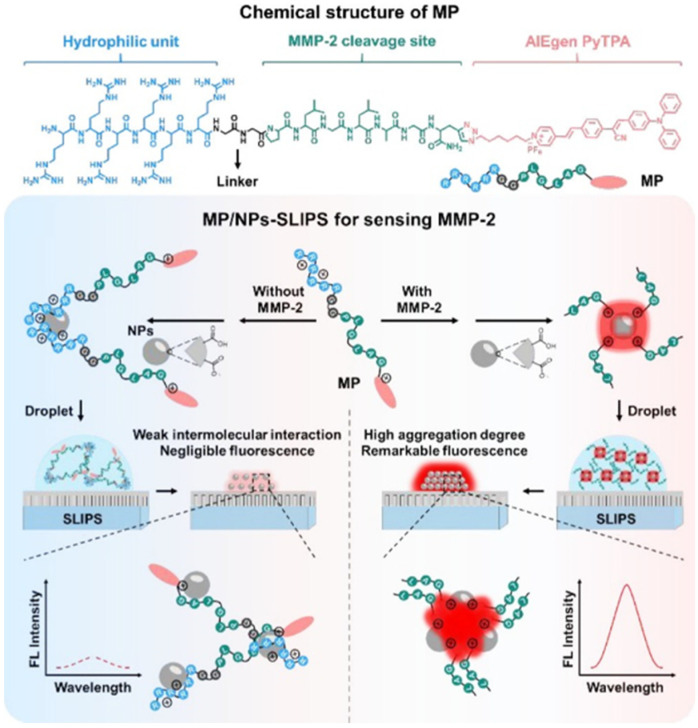
Design of modular peptide-conjugated AIEgen MP and development of the MP/NPs–SLIPS sensing system for sensitively detecting tumor marker MMP-2. Reprinted with permission from ref. [45]. Copyright 2021, American Chemical Society.

**Figure 3 biosensors-12-00953-f003:**
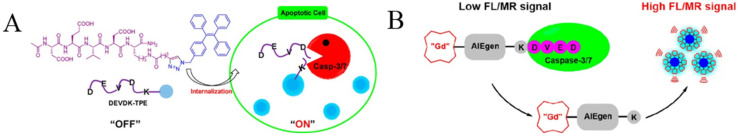
(**A**) Illustration of Ac-DEVDK-TPE for the study of caspase activities. Reprinted with permission from ref. [46]. Copyright 2012, American Chemical Society. (**B**) Schematic illustration of DPBP as a bioprobe for autophagy detection. Reprinted with permission from ref. [49]. Copyright 2019, American Chemical Society.

**Figure 4 biosensors-12-00953-f004:**
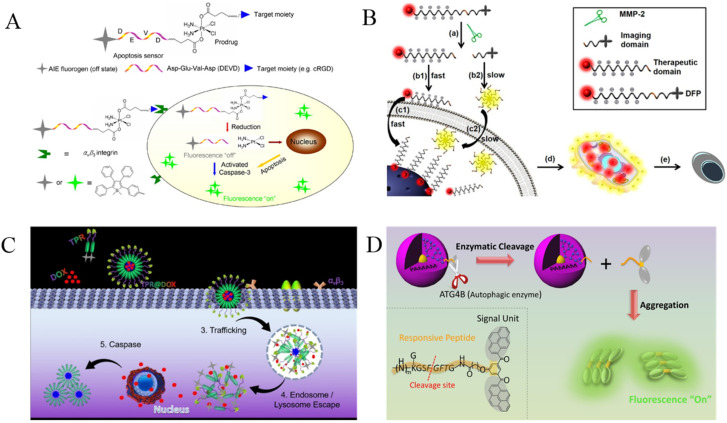
(**A**) Schematic illustration of the targeted theranostic platinum (IV) prodrug with a built-in aggregation-induced emission (AIE) light-up apoptosis sensor for noninvasive, in situ early evaluation of its therapeutic responses. Reprinted with permission from ref. [50]. Copyright 2014, American Chemical Society. (**B**) Schematic illustration showing the probe DFP for rapid drug delivery and drug release tracking in MMP-2 over-expression living cells. Reprinted with permission from ref. [51]. Copyright 2016, American Chemical Society. (**C**) Schematic illustration of TPR@DOX for tumor treatment and apoptosis monitoring. Reprinted with permission from Reference [52]. Copyright 2021, American Chemical Society. (**D**) Caspase-3/7-sensing mechanism of CP1. Reprinted with permission from ref. [53]. Copyright 2017, American Chemical Society.

**Figure 5 biosensors-12-00953-f005:**
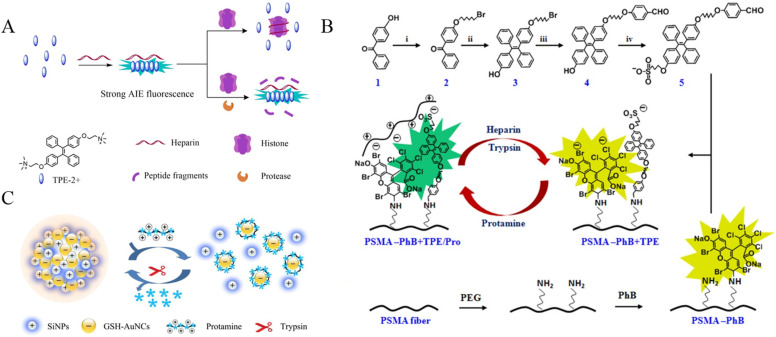
**(A**) Schematic illustration of the fluorometric assay for protease activity. Reprinted with permission from ref. [55]. Copyright 2018, Elsevier. (**B**) Synthetic route of TPE derivatives, schematic drawing of the ratiometric color changes of fibers, and grafting process of PhB on electrospun PSMA fibers. Reprinted with permission from ref. [56]. Copyright 2017, American Chemical Society. (**C**) Schematic illustration of the mechanism of the ratiometric detection of protamine and trypsin based on a nanohybrid probe. Reprinted with permission from ref. [57]. Copyright 2019, Elsevier.

**Figure 6 biosensors-12-00953-f006:**
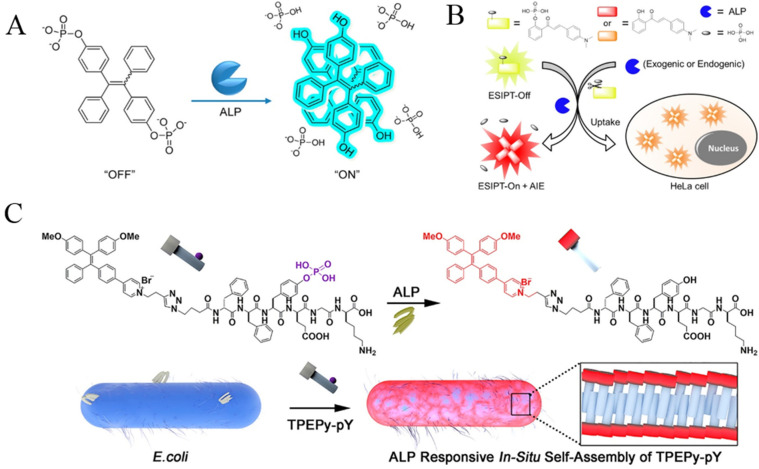
(**A**) Illustration of the design principles of ALP assay with a TPE-2PA probe. Reprinted with permission from ref. [59]. Copyright 2013, American Chemical Society. (**B**) Schematic illustration of HCAP for ALP activity assay in solution and in living cells. Reprinted with permission from ref. [63]. Copyright 2014, American Chemical Society. (**C**) Schematic illustration of the peptide self-assembly-controlled turn-on probe for sensing ALP activity. Reprinted with permission from ref. [65]. Copyright 2020, American Chemical Society.

**Figure 7 biosensors-12-00953-f007:**
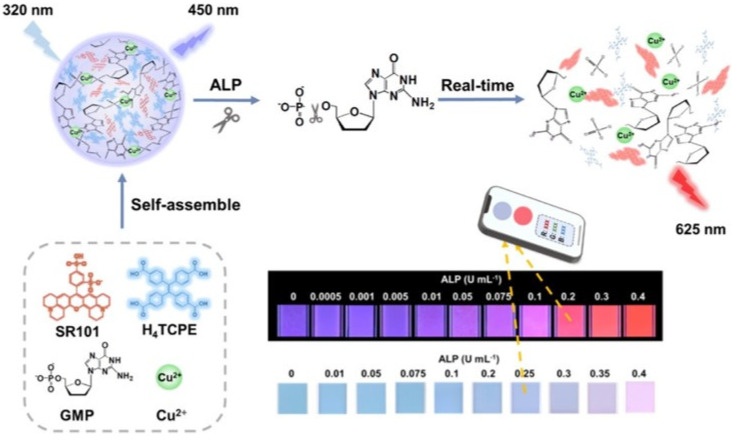
Schematic illustration of the real-time monitoring of ALP activity, based on the ratiometric fluorescence response of H_4_TCPE/SR101/Cu-GMP ICP nanoparticles stemming from the AIE guest and the ACQ guest simultaneously, and its field application for algal bloom warning implanted with a smartphone. Reprinted with permission from ref. [69]. Copyright 2021, American Chemical Society.

**Figure 8 biosensors-12-00953-f008:**
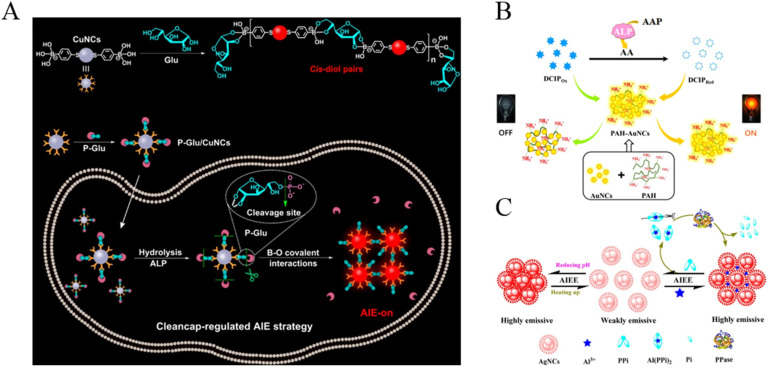
(**A**) Schematic illustration of (top) the proposed mechanism for specific recognition between MBA-stabilized CuNCs and glucose and (bottom) the cleancap-regulated AIE strategy for the imaging of ALP activity. Reprinted with permission from ref. [72]. Copyright 2020, American Chemical Society. (**B**) Schematic illustration of the detection strategy for ALP activity based on FRET. Reprinted with permission from ref. [73]. Copyright 2019, Elsevier. (**C**) Schematic illustration of AIEE-based AgNC nanoswitches in response to multiple stimuli and a detection strategy for PPase activity based on ion-triggered switch. Reprinted with permission from ref. [75]. Copyright 2017, American Chemical Society.

**Figure 9 biosensors-12-00953-f009:**
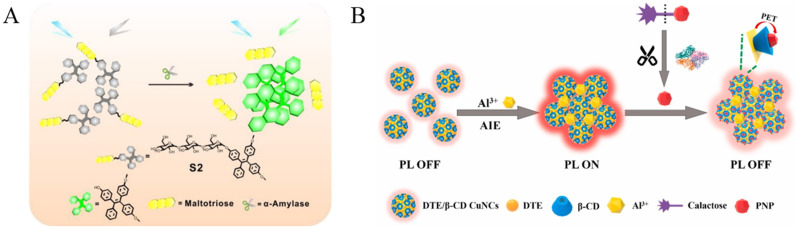
(**A**) Schematic illustration of the sensing mechanisms of the probe S2 in α-amylase activity sensing. Reprinted with permission from ref. [84]. Copyright 2018, American Chemical Society. (**B**) Schematic representation of the strategy for the determination of β-Gal. Reprinted with permission from ref. [87]. Copyright 2020, American Chemical Society.

**Figure 11 biosensors-12-00953-f011:**
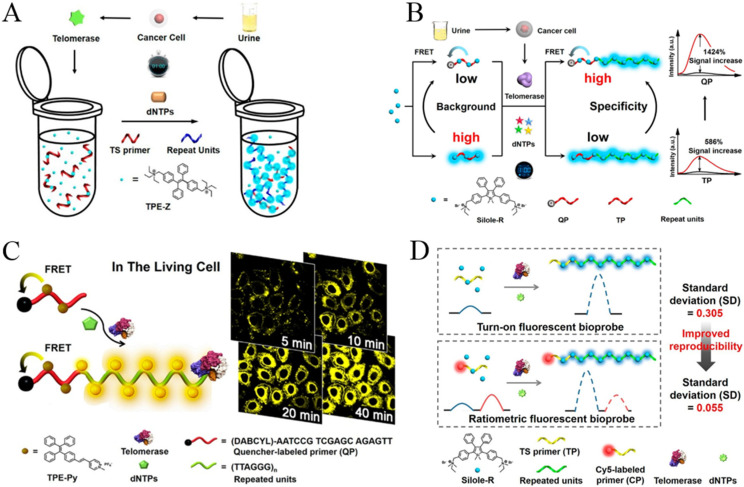
(**A**) Schematic illustration of the AIE-based simple one-pot technique for telomerase activity detection. Reprinted with permission from ref. [97]. Copyright 2015, American Chemical Society. (**B**) Schematic illustration of the quencher group-induced high specificity fluorescence strategy for the detection of telomerase activity. Reprinted with permission from ref. **[98]**. Copyright 2015, American Chemical Society. (**C**) Schematic Illustration of the AIE-based in situ telomerase activity detection and imaging. Reprinted with permission from ref. [99]. Copyright 2016, American Chemical Society. (**D**) Schematic illustration of the ratiometric fluorescent bioprobe for telomerase activity detection. Reprinted with permission from ref. [100]. Copyright 2016, American Chemical Society.

**Table 1 biosensors-12-00953-t001:** Overview on AIE-based methods for the detection and imaging of different proteases.

Targets	Probes	Linear Range	LOD	Ref.
LAP	DPA-TPE-Leu	–	8.9 ng/mL	[39]
LAP	ASSI-Leu	0~0.05 U/mL	2.983 mU/mL	[40]
CKII	TPE-GRRRADDSDDDD	0.1~80 mU/μL	0.05 mU/μL	[41]
chymase	TPETH-2(CFTERD_3_)	0~9.0 ng/mL	0.1 ng/mL	[42]
caspase-3	TPE-GFFYK(DVEDEE-Ac)	0~70 pM	0.54 pM	[43]
Furin	Ac-RVRR-linker-HPQ	Imaging	–	[44]
MMP-2	RRRRRRGGPLGLAGPra(PyTPA)-NH_2_	Imaging	–	[45]
caspase-3/7	Ac-DEVDK-TPE	Imaging	–	[46]
caspase-3	Ac-DEVD-TPS-cRGD	0~80 ng/mL	–	[47]
caspase-3/8	TPETH–DVEDLEHD–TPS	Imaging	–	[48]
caspase-3/7	DOTA-Gd(III)-TPE-KDEVD	Imaging	–	[49]
caspase-3	TPS-DEVD-Pt-cRGD	–	1 pM	[50]
MMP-2	DOX-FCPPs-PyTPE	Imaging	–	[51]
ATG4B	DPBP	Imaging	–	[53]
trypsin	TPE-2+/BSA	1~12.5 mU/mL	1.43 mU/mL	[54]
trypsin	TPE-2+	0.1~1 nM	0.02 nM	[55]
trypsin	PSMA-PhB+TPE/heparin	0~1.5 U/mL	0.02 U/mL	[56]
trypsin	SiNPs@GSH-AuNCs/protamine	0.15~3.0 μg/mL 10~100 ng/mL	0.07 μg/mL 4.50 ng/mL	[57]

Abbreviations: LAP, leucine aminopeptidase; DPA, diphenylamine; Leu, leucine; CKII, casein kinase; MMP-2, matrix metalloproteinase-2; TPS, tetraphenylsilole; PyTPE, tetraphenylethene derivative; DOX, doxorubicin; FCPPs, functionalized cell penetrating peptides; ATG4B, an autophagy-specific enzyme; PSMA, polystyrene-co-maleic anhydride; PhB, phloxine B; SiNPs, silicon nanoparticles; GSH, glutathione; AuNCs, gold nanoclusters.

**Table 2 biosensors-12-00953-t002:** Analytical performances of AIE-based methods for the detection of phosphatases.

Targets	Probes	Linear Range	LOD	Ref.
ALP	TPE-PA	0~0.1 U/mL	18 mU/mL	[58]
ALP	TPE-2PA	3~526 U/L	0.2 U/L	[59]
ALP	TPE-4PA	10~50 mU/mL	–	[60]
ALP	THP	0~200 U/L	1.21228 U/L	[61]
ALP	QMTP	0~1200 U/L	5.36 U/L	[62]
ALP	HCAP	0~150 mU/mL	0.15 mU/mL	[63]
ALP	FAS-P	1~100 U/L	0.6 U/L	[64]
ALP	TPEPy-^D^F^D^F_P_Y^D^EG^D^K	1~10^6^ CFU/mL	6.6 × 10^−3^ U/mL	[65]
ALP	TPE-Py-FpYGpYGpY	0~2 U/mL	–	[66]
ALP	PET-Flu-PO_4_/TPE	0~100 mU/mL	5 mU/mL	[67]
ALP	BSPOTPE-PrS	0~36 mU/mL	28.7 μU/mL	[68]
ALP	H_4_TCPE/SR101/Cu-GMP ICP	0.01~0.1 U/mL	0.0032 U/mL	[69]
ALP	GSH-capped CuNCs	0.5~25 mU/mL	0.15 mU/mL	[70]
ACP	CuNCs	2.2~100 U/L	0.8 U/L	[71]
ALP	P-Glu/CuNCs	0.56~30 U/L	0.17 U/L	[72]
ALP	PAH-AuNCs	0.5~100 U/L	0.2 U/L	[73]
ACP	GSH-AuNCs	0.005~2.4 U/L	0.001 U/L	[74]
PPase	AgNCs	2.1~35 U/L	0.7 U/L	[75]
PKM2	TEPC466	0~20 μg/mL	21.25 ng/mL	[76]

Abbreviations: ALP, alkaline phosphatase; TPE, tetraphenylethylene; PA, phosphate; THP, 2-(benzo[d]thiazol-2-yl)-4-(1,4,5-triphenyl-1H-imidazole-2-yl)phenyl dihydrogen phosphate; QMTP, the conjugate of quinolone–malononitrile and phosphate-modified thiophene; HCAP, phosphorylated conjugate of 2′-hydroxyacetophenone and 4-dimethylaminobenzaldehyde; FAS-P, (E)-2-(((9H-fluoren-9-ylidene) hydrazono)methyl) phenyl dihydrogen phosphate; BSPOTPE, 1,2-Bis[4-(3-sulfonatopropoxyl)phenyl]-1,2-diphenylethene; pY, tyrosine phosphate; PrS, protamine sulphate; H4TCPE, 1,1,2,2-tetra(4-carboxylphenyl)ethylene; SR101, sulforhodamine 101; GMP, guanosine-5-monophosphate; P-Glu, D-glucose 6-phosphate; CuNCs, copper nanoclusters; GSH, glutathione; PAH, polyallylamine hydrochloride; AuNCs, gold nanoclusters; ACP, acid phosphatase; PPase, inorganic pyrophosphatases; PKM2, pyruvate kinase.

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
