# Peer review of "Biosensors for the Detection of Enzymes Based on Aggregation-Induced Emission"

_biosensors, 2022, doi:10.3390/bios12110953_

Round 1
Reviewer 1 Report
Journal: Biosensors
Manuscript ID: 1994572
Title: Biosensors for the detection of enzymes based on aggregation-induced emission
Comments
In this work, the author’s reviewed AIE materials-based biosensing applications such as biomarker detection. The valuable designs of sensing assays based on AIE active materials have been summarized. This review focused AIE based enzyme-responsive detection mechanisms by classifying the biosensors into six main categories. The review is well written. I recommend an improvement based on the comments below.
An important class of PBIs-based AIE materials is missing in the body of the manuscript. In the last paragraph of section 1, on page 2, mention the PBI-based AIE materials used for various Biosnesing assays (https://doi.org/10.1016/j.talanta.2019.01.023) and PDT applications (https://doi.org/10.1039/C8CC09396B) citing these relevant publications.
2. There are some common parameters of biosensors such as selectivity, sensitivity, LOD, linearity, detection range, method design, etc that can be used for the systematic development and comparison of biosensors. Besides tabulated results. The discussion part should comprise the comparison of biosensors based on these parameters explaining the reasons for performance enhancement.
3. Under each heading of enzymes, the review covers, biomarkers detection for diagnosis, monitoring of physiology, and treatment through imaging. The various applications are merged in a single heading. I suggest splitting them into subheadings for the convenience of readers.
4. Carefully look at the language mistakes, grammar, sentence structures, and formatting. Such as Aa3+ at line 368 should be Al3+ etc.
5. I recommend sorting the tables in ascending or decreasing order of any parameter such as LOD etc to make comparison easier.
6. Besides the organic materials, among the nonmaterial, the AIE quantum dots were also used as biosensors applications. I recommend mentioning quantum dots used for AIE-based biosensors. Such as ALP detection by https://doi.org/10.1016/j.talanta.2017.03.063.
This review provides very useful insights into AIE-based biosensors. I recommend accepting it after an improvement based on the comments above.

Author Response
We thank the reviewer for his/her positive comments: In this work, the author’s reviewed AIE materials-based biosensing applications such as biomarker detection. The valuable designs of sensing assays based on AIE active materials have been summarized. This review focused AIE based enzyme-responsive detection mechanisms by classifying the biosensors into six main categories. The review is well written. I recommend an improvement based on the comments below.
Comment 1: An important class of PBIs-based AIE materials is missing in the body of the manuscript. In the last paragraph of section 1, on page 2, mention the PBI-based AIE materials used for various Biosnesing assays (https://doi.org/10.1016/j.talanta.2019.01.023) and PDT applications (https://doi.org/10.1039/C8CC09396B) citing these relevant publications.
Response: We have mentioned the perylene-based AIE materials and cited the references in the revised manuscript.
Comment 2: There are some common parameters of biosensors such as selectivity, sensitivity, LOD, linearity, detection range, method design, etc that can be used for the systematic development and comparison of biosensors. Besides tabulated results. The discussion part should comprise the comparison of biosensors based on these parameters explaining the reasons for performance enhancement.
Response: It is a good comment. We have discussed the performances of different biosensors in every part. The added sentences have been highlighted in red words.
Comment 3: Under each heading of enzymes, the review covers, biomarkers detection for diagnosis, monitoring of physiology, and treatment through imaging. The various applications are merged in a single heading. I suggest splitting them into subheadings for the convenience of readers.
Response: It is a good suggestion. We did not added the subheadings since each aspect only refers to a few papers (1 ~ 3 references). Thus, we discussed the progress by the overall and subsection.
Comment 4: Carefully look at the language mistakes, grammar, sentence structures, and formatting. Such as Aa3+ at line 368 should be Al3+ etc.
Response: We have checked the manuscript carefully and revised the spelling and grammar mistakes.
Comment 5: I recommend sorting the tables in ascending or decreasing order of any parameter such as LOD etc to make comparison easier.
Response: The works in the tables are sorted based on the type of probes and the position of documents in the article.
Comment 6: Besides the organic materials, among the nonmaterial, the AIE quantum dots were also used as biosensors applications. I recommend mentioning quantum dots used for AIE-based biosensors. Such as ALP detection by https://doi.org/10.1016/j.talanta.2017.03.063..
Response: We have mentioned quantum dots-based AIE materials and cited the references in the revised manuscript.
Comment 7: This review provides very useful insights into AIE-based biosensors. I recommend accepting it after an improvement based on the comments above.
Response: We thank the reviewer for his/her positive comment.
Reviewer 2 Report
The review reports on detection of enzymes based on the aggregation-induced emission (AIE) of different fluorogens. The authors give an overview of the topic by classifying the reported works with respect to the type of enzyme to be detected. The review is well written, and the topic is interesting and relevant.
Some comments and suggestions are reported below:
1: A careful proofreading of the manuscript is required.
2: In the Introduction, there are several periods where specific references are missing, e.g., line 27-38, line 39-42.
3: I suggest adding in the Introduction a brief overview of the main detection mechanism based on AIE in order to aid the understanding of the reported literature.
Author Response
We thank the reviewer for his/her positive comments: The review reports on detection of enzymes based on the aggregation-induced emission (AIE) of different fluorogens. The authors give an overview of the topic by classifying the reported works with respect to the type of enzyme to be detected. The review is well written, and the topic is interesting and relevant. Some comments and suggestions are reported below:
Comment 1: A careful proofreading of the manuscript is required.
Response: We have checked the manuscript carefully and revised the spelling and grammar mistakes.
Comment 2: In the Introduction, there are several periods where specific references are missing, e.g., line 27-38, line 39-42.
Response: We have cited several references to support the presentations.
Comment 3: I suggest adding in the Introduction a brief overview of the main detection mechanism based on AIE in order to aid the understanding of the reported literature.
Response: It is a good suggestion. We have added the sentences to discuss the mechanisms of the works involved in this review in Introduction.